# Assessing the Sound and Heat Insulation Characteristics of Layered Nonwoven Composite Structures Composed of Meltblown and Recycled Thermo-Bonded Layers

**DOI:** 10.3390/polym16101391

**Published:** 2024-05-13

**Authors:** Emel Çinçik, Eda Aslan

**Affiliations:** Department of Textile Engineering, Faculty of Engineering, Erciyes University, 38039 Kayseri, Türkiye; edaaslan1207@gmail.com

**Keywords:** sound insulation, heat insulation, meltblown nonwovens, polypropylene, polybutylene terephthalate

## Abstract

Sound and heat insulation are among the most important concerns in modern life and nonwoven composite structures are highly effective in noise reduction and heat insulation. In this study, three layered nonwoven composite structures composed of a recycled polyester (r-Pet)-based thermo-bonded nonwoven outer layer and meltblown nonwovens from Polypropylene (PP) and Polybutylene terephthalate (PBT) as inner layers were formed to provide heat and sound insulation. Fiber fineness and cross-section of the thermo-bonded outer layer, fiber type (PP/PBT), areal weight (100/200 g/m^2^) and process conditions (calendared/non-calendared) of the meltblown inner layer were changed systematically and the influence of these independent variables on thickness, bulk density, air permeability, sound absorption coefficient and thermal resistance of composite structures were analyzed statistically by using Design Expert 13 software. Additionally, the results were compared with composite structures including an electrospun nanofiber web inner layer and with structures without an inner layer. It was concluded that comparable or even better sound absorption values were achieved with the developed nonwoven composites containing meltblown layers compared to nanofiber-included composites and the materials in previous studies.

## 1. Introduction

Concurrent with increasing technological developments and demands for modern life, noise pollution has emerged as an environmental concern, affecting human health and comfort. Noise, characterized as unwanted sound spanning various frequencies, dulls the senses, reduces concentration, induces difficulty in falling asleep, and causes annoyance. Prolonged exposure to excessive noise can lead to health problems such as tinnitus, hearing impairments, neurological issues, and hypertension. Therefore, noise control is essential for industries like building, automotive, and machinery to improve quality of life [1,2,3,4,5,6].

Nonwoven materials present a promising option for sound absorption due to their substantial and intricately designed internal structure, lightweight nature, cost-effectiveness, and recyclability [4,5,6,7]. The porous nonwoven material consists of fibers of varying lengths and incorporates cavities, channels, or interstices, allowing for the oscillation of air molecules. This oscillation leads to frictional losses that causes the conversion of sound or acoustic energy into heat [7,8]. Moreover, the interlocked fibers within nonwovens act as frictional elements, providing resistance to the motion of acoustic waves. When sound waves encounter the fibers of the structure, they induce vibrations in individual fibers. Additionally, a significant portion of the sound energy can either be absorbed through scattering from the fibers or transformed into heat due to internal frictional forces. In summary, the amplitude of sound energy is reduced or damped by momentum loss, frictional loss and temperature fluctuations occurring in the porous and tortuous structures of nonwovens [2,8,9,10,11,12,13]. Thus, sound-absorbing characteristics of nonwovens depend mainly on pore properties such as size, shape, number and they are affected by the fiber’s diameter, fibers orientation and distribution. The nature of the fiber (type, surface characteristics, elasticity), length, fineness, cross-section of the constituent fibers, the fiber orientation, the porosity, pore size and number, thickness, density and areal weights of the nonwovens and many other factors determine the sound absorption characteristics of these structures [2,5,11,12,13,14,15].

The sound-absorbing properties of nonwoven fabrics have undergone extensive investigations. The effect of raw material type on acoustical characteristics of single-layer nonwovens for various kinds of fibers, such as polyester, flax, jute, kapok, chicken feather, cotton-wool, bamboo, banana, waste wool, recycled polyester, recycled cotton, carbonized cotton, activated carbon, and polyamide/polyethylene bicomponent filaments, were investigated in many studies [4,16,17,18,19,20,21,22,23,24,25,26,27,28,29,30,31,32,33,34,35,36] and it was concluded that different types of fibers can be used in nonwoven production for sound insulation. Fibers with different cross sections (round, trilobal, 4DG, hollow, HexaFlower, flat) were evaluated in nonwovens for sound-absorbing properties in numerous studies [36,37,38,39,40,41,42] and it was revealed that cross sections increasing surface area, pore size and thickness also increased the sound absorption coefficient. Some of the investigations focused on the influence of fiber fineness, areal density, thickness, and the density of nonwovens manufactured with different methods like air laying, carding, needle punching, thermal bonding and vertical lapping [4,5,8,43,44,45,46]. The studies demonstrated that thickness, areal weight, porosity and density are influential factors for sound absorption.

The increase in the number of layers within the nonwoven structure leads to an increase in areal density, thickness, and a change in the percentage of the different types of pores and consequently a change in the acoustic properties. Both the intrinsic characteristics and the sequence of layering play a role in the noise reduction efficiency of multilayered fibrous structures comprising distinct layers [47]. Liu et al. [48] developed a simulation model for the sound absorption coefficient of double-layered nonwovens. The sound-absorbing properties of layered bulky nonwovens produced with traditional methods were studied by evaluating process parameters [15,47,48,49,50,51] and all findings including single-layer and layered nonwovens revealed that heavier and denser nonwovens show good acoustic performance at mid- to high-frequency bands, however they suffer from weak absorption at low and middle frequencies up to 2 kHz. Furthermore, it was concluded that low-frequency sound absorption has a direct relationship with thickness and the effective sound absorption of a porous absorber is achieved when the material thickness is about one-tenth of the wavelength of the incident sound. Eventually, less dense and more open structures are favorable in low-frequency sound absorption [14,15,52,53].

The ways to provide higher sound absorption in low frequency bands is to use nonwovens with higher thicknesses to increase the backing air cavity depth of the nonwovens or to increase the friction between pore walls and air molecules by decreasing pore size and/or increasing total surface area. This can be achieved by decreasing fiber diameter in nonwovens. Thinner fibers move more easily than thicker counterparts when encountered with sound waves. Furthermore, a greater quantity of fibers is necessary to achieve the same volume density with fine denier fibers which results in a more tortuous path and higher airflow resistance. These cases lead to an improvement in sound absorption performance [2,8,54]. Thus, nonwovens containing microfibers and nanofibers or microfiber/nanofibers layers provide superior sound absorption in a large frequency range [55,56,57,58,59,60,61,62,63,64,65,66,67,68,69,70,71,72,73,74,75,76,77,78,79,80].

In recent years, electrospun nanofibers have become popular among sound insulation materials due to their very low diameter, higher surface area, and highly porous network of interconnected pores. The nanofiber structures ensure sound absorption by acting as an acoustic resonance membrane. Numerous researchers have explored the acoustic characteristics of materials incorporating nanofibers, whether in the form of a web layer and/or as reinforcement in nonwovens or textile structures [55,56,57,58,59,60,61,62,63,64,65,66,67,68,69,70,71,72,73,74,75,76,77,78,79,80]. These studies focused on the diversity of polymers [6,56,57,58,59,60,61,62,63,64,65,66,67,68,69,70,71,72,73,74,75,76,77,78,79,80] such as Polyvinyl alcohol (PVA), Polyacrylonitrile (PAN), Polyurethane (PU), Polyvinylpyrrolidone (PVP), Polyamide (PA), Polyvinyl Chloride (PVC), Polystyrene (PS), PVA/Polyethylene oxide/Graphene oxide, recycled Polyethylene terephthalate (r-PET), core-shell and hollow shaped PAN/PVA. All the previous studies have demonstrated that nanofiber webs exhibit high sound absorption at low and medium frequencies alone and integrating nanofibrous structures into nonwovens and textile structures without adding extra weight or thickness. On the other hand, the production process of nanofibrous surfaces is challenging, and producing a layer takes a long time. In this respect, surfaces containing fibers with a small diameter such as microfibers can be beneficial for acoustic applications and meltblown nonwovens can serve as an alternative to nanofibers.

The polymer is converted into continuous spun filaments which are later accelerated by hot and fast-flowing air to form low-diameter fiber changing between 1 and 5 µm in the meltblown process. The formed thinner fibers then accumulate on a collector to construct a self-bonded web layer [81]. Meltblown nonwovens can be economically manufactured representing lighter nonwoven fabrics with reduced fiber and pore diameter, and increased surface area. This material offers an effective alternative for sound absorption management compared to the bulky needle-punched nonwoven sound absorbers available commercially or materials including nanofibers [7,11,58,82,83]. Some studies related to sound insulation features of nonwoven composites evaluated the Polypropylene (PP) [11,58,83], Polylactic Acid (PLA) [11,47], Polyethylene terephthalate (PET) [7] and PP/Pet bicomponent fiber [83]-based meltblown layers.

Çelikel and Babaarslan [7] investigated the sound absorption properties of Spunbond/Meltblown/Spunbond (SMS) multilayer nonwoven structures incorporating bicomponent/homocomponent, round/trilobal PET fiber-based spunbond layers and meltblown layers with increasing areal weight. It was concluded that all samples exhibited inadequate sound absorption performance for frequencies up to 3000 Hz. However, at higher frequencies, three-layered nonwoven structures with bicomponent fibers as outer layers demonstrated superior sound absorption performance compared to nonwoven structures incorporating homocomponent fibers. Additionally, increasing the meltblown layer’s areal weight enhanced sound absorption.

Öztürk et al. [58] evaluated the contribution of differently configured SMS cover layers on the sound-absorbing properties of nanofibrous resonant membrane-coated wool-based needle-punched nonwoven composites. The study showed that the addition of a covering material to the layered structure made a positive contribution to the sound absorption property unless its areal density was lower than 60 g/m^2^. The highest sound absorption coefficient was obtained for composite structures having a 300 g/m^2^ meltblown layer at 500 and 1000 Hz frequencies as 30% and 80%, respectively.

The acoustic performance of cotton, polyester, cotton/polyester-blended needle-punched, PP and PLA-blended meltblown nonwovens were compared in the frequency region of 100–1500 Hz by considering one-, two- and three-layered structures [47]. The research showed that the polypropylene microfiber meltblown nonwoven sample exhibited effective sound absorption behavior across the entire frequency range. Utilizing multilayer samples enhanced the sound absorption coefficient, particularly when one of the layers consisted of a thin meltblown nonwoven layer. This improvement could reach up to 50%, especially when the upper layer was composed of finer fibers from a meltblown nonwoven with a low air permeability value.

A noise-reducing sound absorber designed for attachment to textile machine bodies was created by combining PLA meltblown nonwovens with rigid polyurethane foam (PUF) [11]. The study investigated the impact of fiber diameter, air permeability, pore diameter, volume density, and thickness on the sound absorption properties of PLA meltblown nonwoven materials. The findings revealed that a thin sample with low fiber diameter, the smallest pore diameter, high air permeability, and low density demonstrated significant sound absorption characteristics. It was suggested to use PUF covered by a single layer of PLA meltblown instead of PUF covered by PLA meltblown nonwoven layers on both sides for the design of the sound absorber.

Sivri and Haji [82] identified the most efficient medium for sound absorption performance, exploring various arrangements of polypropylene-based needle-punched nonwoven, polypropylene meltblown nonwoven, and hybrid forms, and examining their correlation with thermal conductivity. The composite structure where the meltblown nonwoven with the lowest fiber diameter was placed at the face side of needle-punched nonwoven was found to provide the highest sound absorption coefficient; nonetheless, sound absorption was inadequate for sounds with medium and low frequencies.

The paper of Lee et al. [83] presented the design of a three-layer composite structure for sound-absorbing material comprised of a surface layer with fine meltblown and high-modulus hollow fiber, a middle layer with bulky nonwoven and a bottom layer with meltblown nonwoven layers. The influences of fiber diameter, areal weight, thickness, and air permeability on sound absorption were investigated and it was concluded that bulky meltblown nonwovens were effective in sound insulation, and increasing weight and thickness enhanced the acoustic properties.

As a result of a detailed literature survey, it was concluded that although an extended number of papers have been published about the acoustic properties of nonwoven structures, there has been a lack of systematic research on the sound absorption characteristics of meltblown nonwoven integrated nonwoven structures. Previous studies primarily focused on the areal weight of meltblown nonwovens, neglecting other important parameters, and comparisons with nanofiber web counterparts were also lacking.

In addition to sound insulation requirements, concerns for energy conservation have contributed to the need for economical heat/thermal insulation for buildings, automobiles, aircraft and industrial process equipment and clothing. Low price, low weight, easy production processes and diversity of structural porous properties caused nonwoven materials to be one of the most important products used for heat insulation as well. Previous studies [18,26,30,31,43,44,84,85,86,87,88,89,90,91,92,93,94,95] evaluating nonwovens produced many different techniques which indicated that fiber type, thickness, bulk density, porosity and any factors influencing these structural parameters alter the thermal insulation properties such as thermal resistance or thermal conductivity. Furthermore, the literature review revealed that previous research on thermal insulation primarily addressed single-layer nonwovens, with limited attention given to layered structures, particularly meltblown nonwovens.

This study contributes to the field by addressing the lack of research on the sound absorption and thermal insulation properties of multilayered nonwoven composite structures, particularly those incorporating meltblown nonwovens. Additionally, it fills the gap in the literature by systematically evaluating various parameters about meltblown nonwovens and comparing the results with structures containing nanolayers or no inner layer. In this study, fiber fineness and cross-section of thermo-bonded layer as outer layers, fiber type, areal weight, and production process of meltblown layers as inner layers were systematically changed. The influence of these parameters on sound and heat insulation of the composite structures were statistically analyzed by using Design Expert software to determine the most effective structures with regard to different sound frequencies. Also, the results were compared with the composite structures including a nanoweb layer as the inner layer and composite structures without the inner layer. It was concluded that comparable or even better sound absorption values and similar heat insulation values were achieved with the developed nonwoven composites.

## 2. Materials and Methods

### 2.1. Materials

In light of all the studies discussed in the Introduction section, it has been determined that nonwoven surfaces with different characteristics are effective at different sound frequencies. Therefore, three-layered nonwoven composite structures were constituted by assembling two thermo-bonded nonwoven layers as outer layers and meltblown/nanofiber nonwoven webs as inner layers to enable a single material for addressing different sound frequencies. The inner layers of the composite structure were formed either with meltblown technology by altering raw material, areal weight and process type or with electrospinning technology.

The thermo-bonded nonwovens used for the outer layer were produced from recycled polyester (r-PET) fiber with different fiber fineness and cross-section through carding and thermo-bonding processes, sequentially. The properties of the fibers constituting the thermo-bonded layers are presented in Table 1. As seen from the table, the thermo-bonded layers were formed with 7-denier solid, 7-denier hollow and 12-denier hollow r-Pet fibers to provide sustainability and bicomponent polyester fiber. The bicomponent polyester fiber is composed of a standard polyester center and co-Polyester shell part with a 110 °C melting point. The air gap of the 7-denier hollow r-Pet fiber was larger than that of 12-denier hollow r-Pet fiber.

### 2.2. Methods

#### 2.2.1. Manufacturing of Nonwoven Layers and Layered Nonwoven Composite Structures

The outer thermo-bonded layers were produced by following the carding with the web forming method and thermal bonding with the web bonding method, respectively. The r-PET fibers and bicomponent polyester fibers were blended with an eight-chamber mixer according to the fiber contents given in Table 2 after the application of various preparation processes such as bale opening and opening fibers. In the table, the layers were coded considering the fibers forming the web. For instance, 7DH represents the thermo-bonded outer layer constituted with 80% 7-denier hollow r-PET and 20% bicomponent polyester fiber. Mixed fibers were carded to produce webs with 25 g/m^2^ areal weight and cross-lapped in eight layers to form a web with a 200 g/m^2^ target areal weight. The bonding process of the webs was carried out by the through-air-bonding method in an oven at 200 °C with a band distance of 12 mm and a passage speed set to 50 m/min.

The inner layers of the composite structure were produced with either Polypropylene or Polybutylene Terephthalate as a raw material by using the meltblown method. Also, only one type of electrospun nanofiber web was alternatively used as an inner layer for the purpose of comparison. The raw material, the process parameters (calendered/non-calendered) and the areal weight (100/200 g/m^2^) of meltblown nonwovens were altered and used as independent variables in statistical analyses. The calendering process was performed by using diamond-shaped cylinders at 160 °C for calendered inner layers. The coded names and features of the inner layers are shown in Table 3.

Different nonwoven inner layers with varying properties given in Table 3 were inserted between two thermally bonded nonwovens with different characteristics (Table 2) in order to form three-layered nonwoven composite structures (Figure 1). The layered structures were exposed to heat in an oven without pressure at 160 °C for 10 min to bond the layers to each other.

#### 2.2.2. Testing of Nonwoven Layers and Layered Nonwoven Composite Structure

The individual inner and outer layers and constituted composite structure layers were conditioned according to ISO 139 [96] under standard atmospheric conditions for 24 h before the testing procedure. All the tests were conducted in the standard atmosphere of 20 ± 2 °C and 65 ± 4% humidity. The standard tests were performed to determine the areal density and thickness of the individual layers and composite structures. The areal weights were measured according to the test standard NWSP 130.1 [97] by testing 30 cm × 30 cm of ten samples. A digital thickness gauge (Elastocon EV 07, Brämhult, Sweden) was used to measure the thickness of the inner layers following the NWSP 120.1 [98] test standard. On the other hand, the thickness of the outer layers and composite structures were detected by digital calipers since these structures are voluminous and sensitive to pressure. The bulk density (*d_n_*, g/cm^3^) of the layers and composite structures were calculated using the mean of measured areal weight (*W*, g/m^2^) and thickness (*t*, mm) as follows [99]:(1)$$d_{n}=\frac{W}{1000\times t}$$

The porosity (P; %) of the samples was computed using the bulk density of the nonwoven structure (*d_n_*; g/cm^3^) and density of the fiber forming the structure (*d_f_*; g/cm^3^) as follows [99]. Since the samples were constituted from different fibers, the fiber densities were also calculated based on a weighted average [99]. The densities of r-PET, bicomponent PET, PP, and PBT were considered as 1.35 g/cm^3^, 1.38 g/cm^3^, 0.9 g/cm^3^, and 1.35 g/cm^3^, respectively, during calculations [100,101,102,103,104].
(2)$$P=(1-\frac{d_{n}}{d_{f}})*100$$

Furthermore, due to the different structural parameters, the pore size characteristics of meltblown layers were also determined according to the ASTM E1294 [105] test standard via a capillary flow porometer (PMI, Florham Park, NJ, USA).

The air permeability tests of samples were performed on a digital air permeability tester (Textest FX 3300, Zurich, Switzerland) following the NWSP 070.1 [106] test standard using a test area of 20 cm^2^. The results were expressed as L/m^2^/s by taking into consideration the unit volume of air (l) that passed through 1 m^2^ of material at a pressure difference of 200 Pa in one second. The thermal conductivity of the samples was determined according to ASTM C518 employing a heat flow meter (Thermtest HFM-100, Hanwell, NB, Canada). The thermal resistance (R: m^2^K/W) of the structures was calculated based on thermal conductivity (λ: W/mK) and thickness (*h*: m) as given below [86,90]:(3)$$R=\frac{h}{\lambda }$$

The sound absorption coefficients of multilayered composite nonwoven structures were computed utilizing an impedance tube (Brüel & Kjær 4206 model, Nærum, Denmark) in accordance with the ISO10534-2 [107] and ASTM E1050–08 [108] standards. The prescribed test methodology encompasses the utilization of an impedance tube as displayed in Figure 2. In this setup, a sound source (loudspeaker) is positioned at the left end of the impedance tube, while the sample is situated at the right end. The sound source generates broadband, stationary random sound waves that propagate as plane waves within the tube. The propagation, contact, and reflection processes lead to a standing-wave interference pattern due to the superposition of forward- and backward-traveling waves inside the tube. The measurement involves recording the sound pressure at two fixed locations and computing the complex transfer function using a two-channel digital frequency analyzer. This enables the determination of sound absorption and complex reflection coefficients, as well as the normal acoustic impedance of the material. The applicable frequency range is contingent upon the diameter of the tube and the spacing between the microphone positions. The sound absorption capacity of the constructed samples was assessed across the 50–6300 Hz frequency spectrum by employing both large and small tubes. The large tube is employed for the 50–1600 Hz sound frequency range, whereas the small tube is utilized for the 1600–6300 Hz frequency range. The sample diameters for the large and small tubes are 100 mm and 29 mm, respectively [82,109].

The results derived from the tests were statistically analyzed by using Design Expert software. During analyses, independent variables were chosen as the type of outer layer (O; 7DS, 7DH, 12DH), the raw material of the inner layers (R; PP/PBT), the process parameter of the inner layer (P; C/NC) and the areal weight of inner layer (W; 100/200 g/m^2^). Moreover, the frequency level of sound (F) was used as an additional independent factor while evaluating the sound absorption coefficient property. The dependent factors were thickness, bulk density, air permeability, thermal conductivity, and sound absorption coefficient of multilayered nonwoven composite structures. As a result of statistical analyses, analysis of variance (ANOVA) tables of each composite property were evaluated and the variation of each composite feature with chosen variables was assessed through the graphs formed by the software.

## 3. Results and Discussion

### 3.1. Properties of Individual Layers Forming the Composite Structures

The physical properties such as areal weight, thickness, bulk density, porosity, pore size, air permeability, etc. of each layer are presented in Table 4 and Table 5 for the outer and inner layers, respectively.

### 3.2. Properties of the Nonwoven Composite Structures

The features of layered nonwoven composite structures such as thickness, bulk density, air permeability, sound absorption coefficient and thermal resistance were evaluated with statistical analyses in this section.

#### 3.2.1. Thickness and Bulk Density

The thickness and bulk density are key factors for nonwovens to explain both structural properties and the relation between structure and performance properties. The summarized analysis of variance (ANOVA) tables of thickness and bulk density values belonging to nonwoven composite structures are demonstrated in Table 6. Here, R, W, and P represent the raw material, areal weight and process parameter of the inner layer, respectively. Moreover, O expresses the type of the outer layer.

The parameters in models having *p* values lower than 0.05 are expected to have a statistically significant effect in the 95% confidence interval on thickness and bulk density in this table. Significant effect expresses that the chosen independent variable causes statistically meaningful variation to the dependent variable. The contribution of each factor/model is the ratio of the sum of squares of each factor/model to the sum of squares of the corrected total. The contribution of the model is also named R^2^ (the coefficient of determination). R^2^ is a statistical measure that determines the proportion of variance in the dependent variable that can be explained by the independent variables. R^2^ values of the generated models were determined as 98.28% for thickness and 98.82% for bulk density. This expressed that the chosen factors explain 98.28% and 98.82% of the variation in thickness and bulk density, respectively.

As can be seen from the table, all the factors chosen for the experimental study had significant effects on the thickness and bulk density, except the individual effect of inner layer raw material for bulk density. On the other hand, the binary interaction of raw material with process parameter and triple interaction of raw material with process parameter and areal weight contributed significantly to the bulk density of composites. The higher the F values, the higher the effects of the factors acquired. The contribution of outer layer type on thickness and the effects of inner layer areal weight and outer layer type on bulk density were more elevated than other factors when F values were considered.

The influence of various chosen layer parameters on the thickness of nonwoven composite structures is demonstrated in Figure 3. The variation of thickness with inner layer material and inner layer areal weight is presented in Figure 3a for composites including 7DS outer layer and calendared inner layer. The trend was similar for other inner and outer layers.

The composites including higher areal weighted inner layer had a higher degree of thickness, as expected. A higher number of fibers in the cross-section of inner layers with higher areal weight caused a higher degree of thickness for individual inner layers as also follows from Table 5 which resulted in a composite structure with a higher degree of thickness. Furthermore, as seen from the same table, the inner layers produced from polypropylene (PP) polymer had a higher degree of thickness compared to Polybutylene terephthalate (PBT) counterparts due to the lower density and structural properties of PP polymer [12,101,104]. Since the density of PP is lower, more fibers were needed to achieve the meltblown nonwoven with the same areal weight which led to a bulkier and thicker structure than PBT. Generally, a higher degree of thickness was obtained for PP-included composite structures (Figure 3a,b) except composites containing 12DH outer and 100 g/m^2^ inner layer (Figure 3c). The different trend was attributed to the unevenness property special to nonwoven structures. When the composite with different outer layers was compared with composites formed from non-calendered, 100 g/m^2^ areal weighted meltblown inner layers, it was concluded that the highest thickness was observed for composites constructed from 7DH outer layer followed by 7DS and 12DH outer layer containing composites, sequentially (Figure 3c). This tendency was the same for composites other than ones evaluated in Figure 3c. Similar thickness trends were also acquired for individual outer layers (Table 4) and it was considered that these thickness results were reflected in the composites.

The change in bulk density of structures with a 7DS outer layer, created with inner layers weighing 100 and 200 g/m^2^, is demonstrated in Figure 4a,b based on the inner layer raw material and processing type. Similar trends were observed for composite structures with outer layers of 7DH and 12DH. Although closer bulk density values have been obtained for composites, it was indicated that, with a few exceptions, composites containing PBT-based inner layers had relatively higher density (Figure 4a,b). When examining the bulk density values of individual inner layers, it was observed that samples based on PBT had higher density. This characteristic extended to composite structures as well. The observed differences in the exceptional samples were presumed to originate from regional variations inherent in the structure of each nonwoven layer.

Additionally, from the figures it was determined that samples with calendered inner layers generally had higher bulk densities compared to those with non-calendered ones. The calendering process involves compressing the meltblown nonwoven surfaces with pressure and temperature to ensure bonds, resulting in these nonwoven surfaces becoming tighter, more compact, and denser. This effect of the calendering process also manifested in composite structures. Moreover, as expected, an increase in inner layer weight led to an increase in the bulk density of the samples.

The impact of the outer layer type of layered structures on the bulk density of the structures is illustrated in Figure 4c for samples with a PBT-based inner layer weighing 200 g/m^2^. Similar trends were found for samples with inner layers of other weights and compositions. Upon examination of the figure, it was concluded that the densest structure was achieved with the 12DH outer layer, followed by 7DS and 7DH outer-layered nonwoven composite structures, respectively. A similar ranking was evident in the individual density results of the outer layers, as seen in Table 4. We believe that the outer layer type, identified as the most influential parameter in volumetric density based on the ANOVA table (Table 6), imparts a similar trend to the composite as it was in the individual bulk density of single outer layers.

#### 3.2.2. Air Permeability

Air permeability refers to the ability of air to pass through the fibers and fabric structure and that of the composite structures was determined to help the understanding of the overall structure and porosity of layered nonwoven composite structures. The most suitable model explaining the air permeability of layered nonwoven composite structures with different featured layers has been determined as a modified cubic model through statistical analysis, and the ANOVA table for this model is shown in Table 7.

As follows from the table, the effect of all the chosen factors related to nonwoven composites and their binary and triple interactions on air permeability were statistically significant. It is evident that the created model explained 99.29% of the variation in air permeability of the layered structure. The most influential parameter contributing to the air permeability of the layered structure was found to be the processing type of the inner layer (61.65%), followed by the areal weight of the inner layer contributing 20.96%, and the interaction of the processing type with the inner layer raw material with a contribution of 5.89%.

The variation in air permeability with different outer and inner layers for chosen layer factors is discussed in Figure 5. When the air permeability values of composites with different outer layers and inner layers with different raw materials were examined from the figure, it was determined that the highest air permeability for all samples was achieved in structures with the outer layer of 12DH, followed by structures with the outer layer of 7DH and 7DS. A similar ranking existed in the individual air permeability values of the outer layers, (Table 4), suggesting that the characteristics of individual layers also influenced the layered structure. Due to the higher linear density of 12DH fibers, there must be fewer fibers in the cross-section to achieve the same areal weight on a nonwoven surface, creating larger and more spaces and allowing air passage between thicker fibers. This situation can be seen in Figure 6 where the surface images of the outer layers are presented. Additionally, considering the lower thickness of this layer, it was estimated that samples obtained from this layer have higher air permeability. In the outer layer obtained with 7DH fibers, it was concluded that, despite its high thickness, a looser and more porous nonwoven surface was obtained due to the hollow structure of the fibers inside, leading to higher air permeability.

Additionally, it was observed that composites containing PBT-based meltblown nonwoven surfaces as the inner layer had slightly higher air permeability than PP-based counterparts, except for structures with 200 g/m^2^ non-calendered inner layers. A similar tendency was also observed for individual inner layers (Table 5). The most crucial parameters known to influence the air permeability property are pore size and thickness. Furthermore, increased thickness causes the air permeability to decrease while increased pore size and porosity lead to an increase in air permeability. Based on the calculated porosity and measured pore size values (Table 5), it was revealed that calendared PBT-based meltblown nonwovens were more porous and had larger pores, while non-calendered PP-based meltblown nonwovens were more porous and had larger pores.

Moreover, the structural twisted fibers observed on the surface of non-calendered PBT meltblown nonwovens with 200 g/m^2^ (Figure 7) were assumed to cover the surface of the layer which resulted in lower air permeability. On the other hand, despite higher porosity and larger pores of PP-based non calendered 100 g/m^2^ meltblown inner layers, lower air permeability was obtained for composites containing this inner layer. This phenomenon was considered to arise from the higher thickness exhibited by PP-based meltblown nonwovens. As mentioned in Table 5, the thickness difference between the mentioned inner layers was substantial and thus caused composites including non-calendered PBT meltblown with 100 g/m^2^ to have slightly higher air permeability.

To illustrate the impact of the process type of meltblown nonwovens used as an inner layer on the air permeability of composite structures, the air permeability values of samples containing 100 g/m^2^ PP-based inner layers are presented graphically in Figure 8. As expected, the air permeability of composite structures with calendared inner layers was lower compared to those without calendaring. This result was valid for samples with inner layers containing different areal weights and raw materials. The calendaring process involves passing meltblown nonwoven surfaces through hot rollers under pressure, tightly bonding the fibers to each other, consequently causing a reduction in pores and air permeability.

The air permeability values for composites with non-calendered PBT inner layers are displayed in Figure 9 to elucidate the influence of the areal weight of the inner layer on the air permeability of layered structures. Similar outcomes were observed for all composites containing all inner layers. As depicted, an increase in the areal weight of the inner layer led to a reduction in the air permeability of the layered composite structure. The number of fibers resisting air passage in the cross-section rose with an increase in areal weight, consequently resulting in decreased air permeability.

#### 3.2.3. Sound Absorption Coefficient

The sound absorption coefficient is a measure quantifying the extent to which a material or surface absorbs sound energy at a specific frequency, indicating the proportion of incident sound energy that is absorbed rather than reflected or transmitted. It is typically expressed as a dimensionless value between 0 and 1, where 0 represents total sound reflection and 1 represents total sound absorption. The sound absorption coefficients of developed layered nonwoven composites have been measured for 19 frequencies within the range of 100–6300 Hz. The frequency values (F), where sound absorption coefficients were determined, have been treated as an additional independent variable in the statistical analysis to generate a model for elucidating sound absorption properties of the nonwoven composite structures. The ANOVA table of sound absorption coefficient of multilayered nonwoven composites for varying frequencies is evaluated in Table 8.

As indicated in the ANOVA table (Table 8), the most influential factor affecting the sound absorption behavior was found as the sound frequency with a contribution of 42.45%. The R^2^ value of the mentioned model was determined as 94.79% which expresses that the areal weight of the inner layer (W), the raw material of the inner layer (R), the process type of the inner layer (P), outer layer type (O) and sound frequencies (F) explain the 94.79% of the variation in the sound absorption coefficient of the composite structures. All the chosen factors handled were found to have significant effects on the sound absorption coefficient except the raw material of the inner layer. On the other hand, the impact of binary interactions of raw material and sound frequency were significant.

The variation in sound absorption coefficients of multilayered nonwoven composite structures with sound frequencies and raw materials of the inner layer are depicted in Figure 10 for 200 g/m^2^ areal weighted calendered and non-calendered inner layers and 7DH outer layer. In consideration of the similar results of other layers, these two graphs were provided here as exemplary instances. As illustrated from the graphs, an increase in the sound absorption coefficient was indicated with increasing sound frequencies. The observations revealed that composites with calendered inner layers were effective at low and moderate sound frequencies, whereas those without calendared inner layers demonstrated efficacy in higher frequencies. A maximum sound absorption coefficient of 0.46 was obtained for 630 Hz, 0.71 for 800 Hz, and 0.74 for 1000 Hz sound frequencies, respectively. These results for low frequencies were higher than the sound absorption coefficients derived from previous studies conducted with nanofiber layers [56,57,58,59,60,61,62,63,64,65,66,67,68,69,70,71,72,73,74,75,76,77,78,79,80]. Moreover, 0.77–0.98 sound absorption values were also acquired for moderate sound frequencies (1250–3150 Hz) whereas 0.99–1 sound absorption values were provided for high frequencies (4000–6300 Hz) with developed nonwoven composite structures. Furthermore, these results were considerably higher than the results of previous studies [8,9,10,17,18,19,20,21,22,23,24,25,26,27,28,29,30,31,32,33,34,35,36,37,38,39,40,41,42,43,46,47,48,49,82,83,84].

According to the generated model, the impact of the selected structural layer factors seemed to be minimal in comparison to the influence of the sound frequency factor. This was attributed to the markedly distinct behaviors of the developed layered structures in response to different sound frequencies. Since sound absorption coefficients exhibited greater variation at different frequencies, the effect of sound frequencies was believed to overshadow the influence of the selected parameters. For this reason, the sound frequencies were classified as low, medium, and high considering the previous studies [2,11,12,14] and the average sound absorption coefficient for these frequencies was calculated. These average values are also called the noise reduction coefficient (NRC) and were determined for low (100–1000 Hz), medium (1250–3000 Hz), and high (4000–6300) sound frequencies using the following formula [11]:(4)$$NRC=\frac{\sum_{i}^{n}\alpha_{i}}{n}$$

Here, α_i_ expresses the sound absorption coefficient for the first sound frequency and n presents the number of sound frequencies where tests were conducted [11]. For instance, the NRC value for low frequency was calculated by considering the average of α_100_, α_125_, α_160_, α_200_, α_250_, α_315_, α_500_, α_630_, α_800_, and α_1000_. Statistical analyses were performed again for NRC values of high, medium, and low frequencies as dependent factors by taking into account the chosen layer factors such as R, W, O, and P. The summarized ANOVA tables for the mentioned analyses are illustrated in Table 9.

When the table was examined, it was observed that the best models for describing the NRC of the nonwoven composites in low-, medium-, and high-frequency sounds were determined as modified 2FI models containing the main factors (R, W, O, P) and the binary (for example RW, WO, WP) and ternary (WPO) interactions of the factors. The R^2^ values of the models were determined as 93.25%, 95.43%, and 95.39% for low, medium and high frequencies, respectively. Even though the individual effect of all model parameters on noise reduction properties of nonwoven composite structures was significant for all sound frequencies, the raw material of the inner layer in low frequency and outer layer type in high frequency had no individually significant impact. The results revealed that the most influential factors were the areal weight of the inner layer with a 52.82% contribution and the process type of the inner layer with a 16.42% contribution to noise reduction coefficient in low frequency. In the case of mid-range frequencies, the most influential factors on *NRC* were the process type (28.87%) and the areal weight (26.66%) of the inner layer, whereas, at higher frequencies, it was observed that the process type of the inner layer (42.39%), the interaction between processing type and raw material of inner layer (RP) and the interaction of process type and areal weight of inner layer (WP) had the most impact.

The graphs obtained as a result of statistical analysis by software which presented the effects of outer layer type and process type of inner layer on noise reduction coefficient are demonstrated in Figure 11 for low, medium and high sound frequencies. Here, the results of nonwoven composites with PBT-based, 200 g/m^2^ areal weighted and calendered/non-calendered inner layers were demonstrated to display the relation. Similar results were also obtained for composites including other inner layers. As follows from the figures, although the noise reduction coefficients were approximately similar, it is observed that composite structures with a 7DH outer layer provided slightly better sound absorption in low- and mid-frequency ranges (Figure 11a,b). On the other hand, the influence of the outer layer on NRC was negligible for high sound frequencies (Figure 11c). The *p*-value of the outer layer factor was insignificant as indicated in the ANOVA table (Table 9). Samples containing 7DH outer layers exhibited both greater thickness and a more voluminous structure due to the presence of hollow fiber content, consequently yielding higher porosity values (Table 4). For these reasons, it is believed that they provide better sound insulation, thanks to the increased air voids they contain. It has been concluded that these air voids, as indicated in previous studies [2,8,9,10,11,12,13], facilitated the vibration of air molecules for sound attenuation and contributed to the reduction of sound energy. Additionally, they provided the space necessary for the vibration of fibers during the reduction of sound energy.

The influences of process type of inner layers on the NRC of composites could be also evaluated from Figure 11. The NRC of the composites including the calendered inner layer was found to be higher than those that had non-calendered inner layers in low and mid sound frequencies (Figure 11a,b). Conversely, the NRC was higher for composites constructed from non-calendered inner layers than composites formed with calendered inner layers in high sound frequencies (Figure 11c). The tendency was similar for the composites other than those shown in these graphs.

When contemplating the formula of sound waves (λ = c/f; λ: sound wave length-m, c: sound speed-m/s, f: sound frequency-Hz [9,110]); it is observed that frequency is inversely proportional to the wavelength of the sound wave. Therefore, the waves of sound are bigger for lower sound frequencies and vice versa. The fibers adhere to each other more effectively in calendered inner layers due to the effects of heat and pressure, resulting in a denser, tighter, and stronger structure with fewer air voids within. It was concluded that the presence of bonds between the fibers enabled better resistance against larger sound waves. As reported in the literature, resonant-type sound absorbers similar to calendered layers in our study were preferred instead of porous structures in low sound frequencies [9,12,16,41,58]. For high sound frequencies, the wavelength decreases, and the sound waves become denser in number. In this scenario, it was believed that the thick, voluminous and tortuous structure with air voids of non-calendared inner layers provided a favorable environment for attenuating such sound waves. These results were in agreement with previous studies [9,12,16,41].

Figure 12 explains the variation of the NRC with the areal weight of the inner layer located in nonwoven composite structures for low and high sound frequencies. Graphs were formed for composites consisting of PP-based, calendered/non-calendered inner layers and 7DH outer layers, but the same trend was observed for other composite structures. An increase in the areal weight of both calendered and non-calendered inner layers of composite structures led to a rise in NRC when the frequency of sound was in the low- and mid-ranges (Figure 10a). The increasing areal weight of the inner layer caused an increase in the number of fibers in the cross-section. Since the wavelength was high at low and mid frequencies, it was presumed that the increase in the number of fibers interacting with sound waves resulted in greater sound absorption. Furthermore, increasing areal weight means also an increase in thickness which yields higher NRC. Several previous studies also support these results [4,5,8,43,44,45,46].

Regarding the high frequencies, increasing the areal weight of the inner layers in composite structures delivered statistically insignificant differences for composites including non-calendered inner layers, and the NRC values were kept constant although the areal weight of the inner layers increased. However, decreasing NRC values were obtained with ascending inner layer areal weight for the calendered inner layer containing composites. In high frequencies where the wavelength is small and dense, air voids are effective in sound absorption and it would be reasonable to expect higher NRC values with highly porous structures with numerous small air voids. The bulk density and porosity of the non-calendered inner layers with different areal weights were approximately the same whereas the pore size of 100 g/m^2^ is higher as shown in Table 5. Accordingly, the NRC has remained constant. In calendered counterparts, the number and size of pores decreased with increased areal weight and increased the number of fibers in the cross-section due to the calendaring process. It was estimated that with an increase in areal weight, the pores that attenuated sound waves became smaller and the number of pores decreased, consequently this led to a decrease in sound absorption.

The variation between the raw material of the inner layer and NRC is depicted in Figure 13 for different inner layer process types and sound frequencies. The graphs were constituted for composite structures comprising a 7DH outer layer and 200 g/m^2^ inner layer and are shown here, but the trend was the same for others. As seen from the figure, the alteration of the raw material led to a negligible change in NRC at low and mid sound frequencies for composites including both calendered and non-calendered inner layers and at high frequencies for composites containing non-calendered inner layers. This effect was also visible in the graphs containing all sound frequencies (Figure 10). This situation can arise from the insignificant effect of the individual raw material factor for low frequencies displayed in the ANOVA table (Table 9). Also, when the table was examined, it could be inferred that the contribution of raw material individually was lower for mid frequencies (3.81%) and the contribution of singular raw material factor reached up to 10.95% for high frequencies. The porosity values of PP and PBT-based non-calendered inner layers were closer (Table 5) and this result was assumed to be caused by these closer values and the twisted fibers on the surface of PBT non-calendered inner layers (Figure 7) that were believed to eliminate the higher thickness effect and other superior properties of PP non-calendered inner layers.

Considering the calendered inner layers for high sound frequencies (Figure 13c), using PBT-based calendered inner layers had a slightly beneficial effect on the NRC of nonwoven composite structures, as also inferred clearly from Figure 8 for all sound frequencies. This result was attributed to the higher elasticity and resilience properties of PBT fibers compared to PP counterparts (Table 5) which yielded the damping of smaller and more frequent sound waves (high sound frequencies). Furthermore, higher porosity and pore size features of PBT-based calendered inner layers also resulted in higher NRC values for these inner layers.

Figure 14 demonstrates the comparison of NRC values of the nonwoven composite structures developed in this study with both the composite structures with nanofiber inner layers (Table 5) and without inner layers for low (Figure 14a), moderate (Figure 14b) and high (Figure 14c) sound frequencies. The composite structure coded as without an inner layer was formed with two layers of thermo-bonded outer layers whereas nano-coding was constructed by two thermo-bonded outer layers and a nanofiber-containing inner layer.

As indicated in the figures, two layers of thermo-bonded nonwoven composite structures (without an inner layer) exhibited the lowest noise reduction values for all sound frequencies, and adding an inner layer enhanced the sound absorption accordingly. Nanofiber layer-containing composites were found to have slightly lower noise reduction coefficients compared to the meltblown layer-containing counterparts for low and moderate sound frequencies. On the contrary, composite materials containing nanofibers have demonstrated sound absorption properties comparable to those of meltblown competitors at high sound frequencies. As a result of this study, it can be concluded that similar or even higher sound insulation can be achieved with composites formed by using meltblown nonwoven layers compared to nanofiber layers. Considering that meltblown nonwovens can be produced more rapidly, economically, effectively, and easily compared to nanofiber surfaces, it can be seen that the use of meltblown layers will be more advantageous.

#### 3.2.4. Thermal Resistance

Thermal insulation for textile materials and nonwoven structures is often expressed in terms of thermal resistance, which quantifies the ability of a material to resist the flow of heat. The thermal resistance value (R) is a fundamental metric used in the field of insulation to evaluate and compare different materials’ effectiveness in impeding heat transfer. It can be stated that the higher the thermal resistance, the better the insulation provided by the nonwoven structures [111,112]. Table 10 presents the ANOVA results of the thermal resistance of nonwoven composite structures.

Upon reviewing the ANOVA table, it can be seen that 88.94% of the variation in thermal resistance of the nonwoven composite structures could be explained with the chosen parameters in this study (R^2^ = 88.94%). The individual effects of areal weight and process type of the inner layer were found to have a meaningful effect on thermal resistance; on the contrary, raw material of the inner layer had no significant effect. The outer layer type (60.13%) was determined to be the major contributor to the thermal resistance of the composites which denoted that the thermal resistance of the generated composite structures mainly depended on the outer layer type. Furthermore, despite the insignificant individual influence of raw material and process type of the inner layer, the triple effect of RWO and WOP had a meaningful impact on the thermal resistance of the composites.

In general, the type of fiber, thickness, density, and porosity of textile products and nonwovens, which also determines the amount of trapped air within them, play an important role in their thermal resistance properties. Additionally, nonwovens containing fibers with lower thermal conductivity exhibit higher thermal resistance. According to the literature data obtained, the thermal conductivity of PP polymer ranges from 0.17 to 0.22 W/mK [101,113], while the thermal conductivity of PBT polymer ranges from 0.17 to 0.23 W/mK [114,115]. As seen, the thermal conductivity properties of polymers forming the intermediate layers are very close to each other, so the raw material type was not found to be effective as a main factor. However, the parameters such as thickness, density, and porosity also changed according to raw material type; this factor is thought to have an indirect effect on the thermal resistance properties of layered structures.

The thickness, density, and porosity of the composite structures are considered to be prominent in the thermal resistance properties of porous materials. The more voluminous the nonwovens are containing more air voids, the higher their thermal resistance properties, due to the lower thermal conductivity of air [111,112]. Since the air trapped in the structure was an important parameter for thermal resistance, the outer layer which had air containing larger pores assumed to lead the outer layer to be the most influential parameter.

The effect of inner layer areal weight on the thermal resistance of composites with different outer and inner layers is discussed in Figure 15. It was observed that the nonwoven composites with 7DH outer layers exhibited the highest thermal resistance values compared to composites with 7DS and 12DH layers. In addition, except for some composites, 7DS and 12DH outer layers generally followed 7DH including composites in terms of thermal resistance, respectively. As mentioned in thickness part, although single 7DS and 12DH outer layers had different thickness values (Table 4), the thickness of the composites including these outer layers were similar (Figure 3c) and accordingly the exceptions were thought to be derived from the regional variations in all layers forming the three-layered composite. Also, the difference in thermal resistance trend for the exceptional composites containing PBT non-calendered inner layers (Figure 15b) may have originated from the twisted fibers densely seen on the surface of this inner layer (Figure 7). The higher thermal resistance of the composites containing 7DH was attributed to hollow fibers constituting the outer layer and thicker and more voluminous structure than other outer layers. Despite the hollow cross-section of fibers in 12DH outer layers, generally, the thermal resistance of composites containing this outer layer was found to be the lowest due to the lower air gap area in the center of the fiber (Table 1) and lower thickness of nonwoven. It was concluded that the larger air voids formed in the structure of the 12DH outer layer because of the thicker fiber diameter caused thermal loss through the thinner thickness of the nonwoven path and eventually lead to lower thermal resistance.

In general, it was determined that an increase in the areal weight of the non-calendered inner layer (Figure 15a,b) had a positive effect on increasing thermal resistance, except for composites including the 12DH outer layer and non-calendered PBT-based inner layer. This exception was assumed to be caused by the twisted fiber regions on the non-calendered PBT inner layer surface which covered the large air gaps in the 12DH outer layer. An increase in areal weight lead to an increase in thickness along with an increase in the number of fibers in the cross-section and a decrease in air gaps within the cross-section. As the non-calendered inner layers were bulkier and thicker, it was estimated that the increase in weight had a supportive effect on the thermal resistance of the composites including these inner layers.

When the influence of areal weight in composites including the calendered inner layer was examined (Figure 15c,d), significant alteration in thermal resistance was not identified with variations in the areal weight of the calendered inner layer. The increase in areal weight yielded an increase in the amount of fiber in the cross-section in calendered inner layers which had a positive effect on thermal resistance, but the increase in thickness was restricted by the calendaring process. The thicknesses of calendered 100 g/m^2^ and 200 g/m^2^ inner layers were closer. Also, the pore size of the calendered inner layer decreased by increasing the areal weight due to the increasing amount of fiber in the cross-section (Table 5) and this influenced the thermal resistance negatively. On the other hand, the diamond-shaped pattern was generated on the surface of the inner layer thanks to the calendaring process and the size of this pattern was different (Figure 16) because of the different polymer properties of PP and PBT, although the same calendaring roller was used. It was thought that these diamond-shaped calendered regions formed air pores between the layers by the layering process and these pores affected the thermal resistance positively. Thus, the interactions of all these factors were assumed to lead to different trends of inner layer areal weight for these composites.

The influence of the process type of the inner layer on the thermal resistance of nonwoven composites with a PBT-based inner layer and 7DH outer layer is evaluated in Figure 17. A similar trend was determined for other counterparts. As indicated from the figure, the higher thermal resistance values were obtained with a non-calendered inner layer for 100 g/m^2^ areal weighted inner layers. The thicker, bulkier non-calendered inner layers caused thicker composites and also the air pores inside the non-calendered inner layer had an enhancing effect on thermal resistance. However, the thermal resistance of composites was kept constant for 200 g/m^2^ areal weighted inner layers despite different process types (Figure 17). The closer thickness values were acquired for composites with 200 g/m^2^ areal weighted calendered and non-calendered inner layers (Table 5) and the air voids were reduced due to the calendaring effect. On the other hand, the diamond-shaped calendaring pattern was formed which created air voids between the layers. Therefore, it was estimated that the thermal conductivity values of calendared and non-calendared samples were closer as a result of these complicated effects.

If the overall thermal resistance results of generated multilayered nonwoven composites were evaluated, it can be indicated that the thermal resistance results changing between 0.519 and 0.697 m^2^K/W were obtained while thermal conductivity values were between 0.048 and 0.060 W/mK. The highest thermal resistance with the created layers is found to be 0.69665 m^2^K/W, and this value was achieved with a sample having a 7DH outer layer and PP, non-calendered inner layer with the areal weight of 200 g/m^2^. According to the Turkish Standard of TS 825 [116], the materials exhibiting thermal conductivity values smaller than 0.065 W/mK could be considered as heat insulation materials. Therefore, all the layered nonwoven composites formed in this study could be used as heat insulation materials. Also, when compared with the widely used heat insulation materials [117,118], the generated composites can offer a competitive advantage concerning thermal conductivity and cost due to the use of recycled PET outer layers.

## 4. Conclusions

As a result of the experimental study addressing the influence of outer layer type and inner layer properties, such as raw material type, areal weight and process type on thickness, bulk density, air permeability, sound absorption coefficient and thermal resistance of multilayered nonwoven composite structures, it was concluded that the chosen parameters had significant effects on the investigated properties of composites. Furthermore, higher R^2^ values were obtained for the statistical models to explain the thickness, bulk density, air permeability, sound absorption coefficient and thermal resistance of generated nonwoven composites.

Regarding the noise reduction for low sound frequencies (100–1000 Hz) and moderate sound frequencies (1250–3150 Hz), the most influential factors were determined as the areal weight of the inner layer and the process type of the inner layer. In the case of higher frequencies, it was observed that the process type of the inner layer (42.39%), the interaction between processing type and raw material of the inner layer (RP) and the interaction of process type and areal weight of inner layer (WP) had the highest impact on NRC. A maximum sound absorption coefficient of 0.46 was obtained for 630 Hz, 0.71 for 800 Hz, and 0.74 Hz for 1000 Hz sound frequencies, respectively. Moreover, 0.77–0.98 sound absorption values were also acquired for moderate sound frequencies (1250–3150 Hz), whereas 0.99–1 sound absorption values were provided for high frequencies (4000–6300 Hz) with the developed nonwoven composite structures. Accordingly, considerably higher sound absorption coefficients were obtained with the developed nonwoven composites compared to previous studies.

Nanofiber layer-containing composites were found to have slightly lower noise reduction coefficients compared to the meltblown layer-containing counterparts for low and moderate sound frequencies, but similar results were obtained for high sound frequencies with composites composed of a meltblown inner layer. Therefore, it can be concluded that similar or even higher sound insulation can be achieved with composites formed by using meltblown inner layers compared to nanofiber inner layers including counterparts. Moreover, if the quick, efficient, economical and easy manufacturing processes of meltblown nonwovens compared to nanofiber layers is considered, using meltblown layers should clearly be beneficial.

When the thermal resistance and conductivity results were assessed, the highest thermal resistance and accordingly lowest thermal conductivity value was acquired with a composite having a 7DH outer layer and PP, non-calendered inner layer with an areal weight of 200 g/m^2^. Thermal conductivity values between 0.048 and 0.060 W/mK were determined with developed multilayered nonwoven composites, which make these materials competitors with widely used insulation materials. Since the outer layer of the developed composites were formed with recycled PET, the composites were found to be cheaper compared to existing counterparts in the market. Furthermore, the contribution to environmental preservation was provided by using sustainable r-Pet fibers in the bigger part of composites. As a result, considering sound and heat insulation and price and sustainability, the developed multilayered nonwoven composites can be used as insulation materials offering a competitive advantage in required areas.

## Figures and Tables

**Figure 1 polymers-16-01391-f001:**
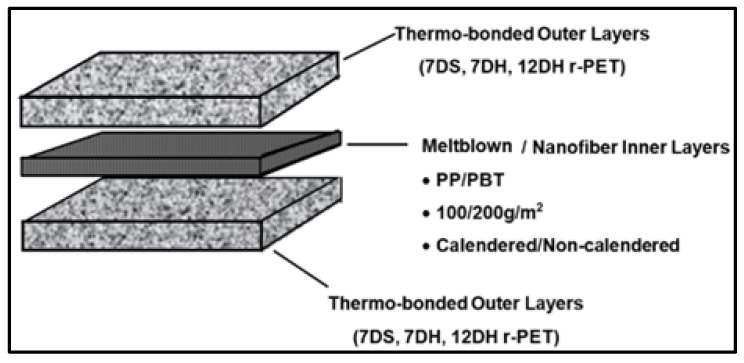
Structure of nonwoven composites.

**Figure 2 polymers-16-01391-f002:**
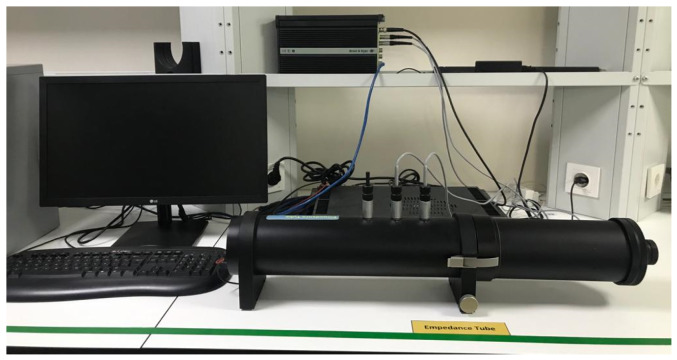
Dual microphone impedance tube device.

**Figure 3 polymers-16-01391-f003:**
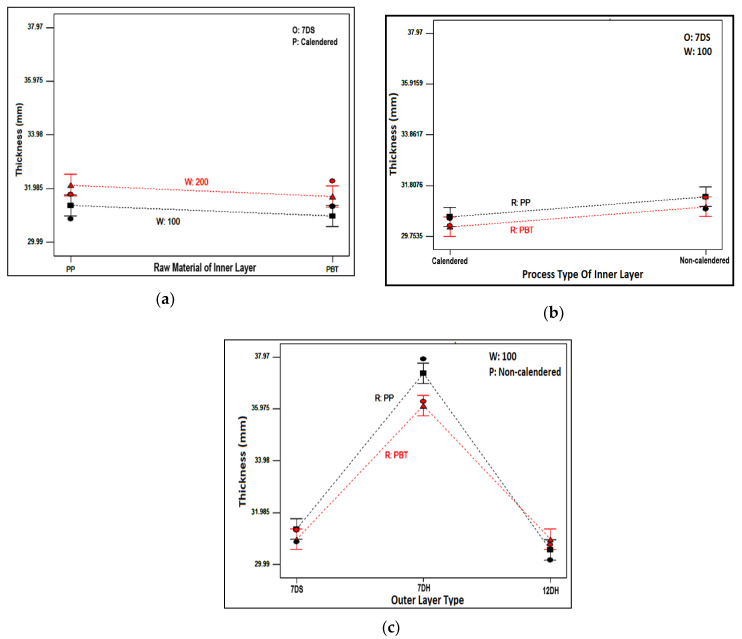
Relation between chosen inner and outer layer parameters and thickness of nonwoven composite structures. (**a**) Relation between areal weight and raw material type of inner layer with composite thickness; (**b**) relation between raw material and process type of inner layer with composite thickness; (**c**) relation between raw material of inner layer and outer layer type with composite thickness.

**Figure 4 polymers-16-01391-f004:**
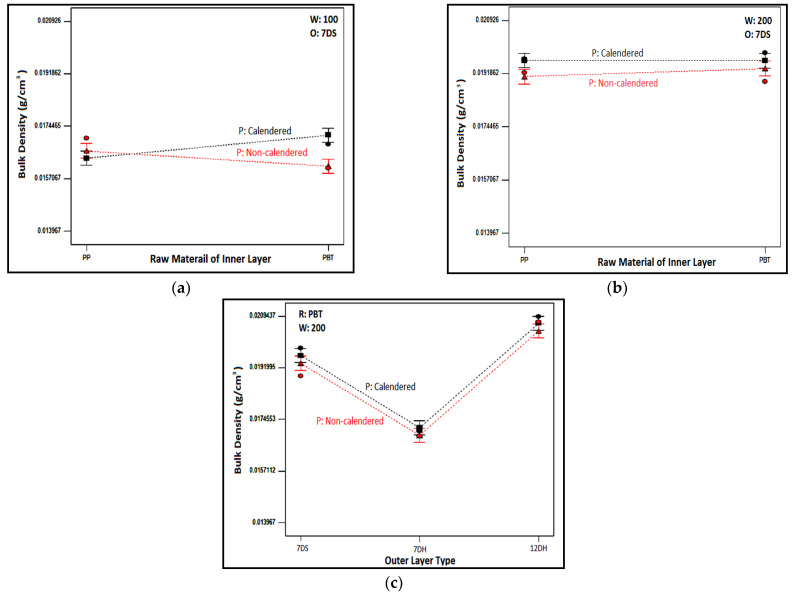
Relation between chosen inner and outer layer parameters and bulk density of nonwoven composite structures. (**a**) Relation between raw material type and process type of inner layer with composite bulk density for composites including 100 g/m^2^ inner layer; (**b**) relation between raw material and process type of inner layer with composite bulk density for composites including 200 g/m^2^ inner layer; (**c**) relation between process type of inner layer and outer layer type with composite bulk density.

**Figure 5 polymers-16-01391-f005:**
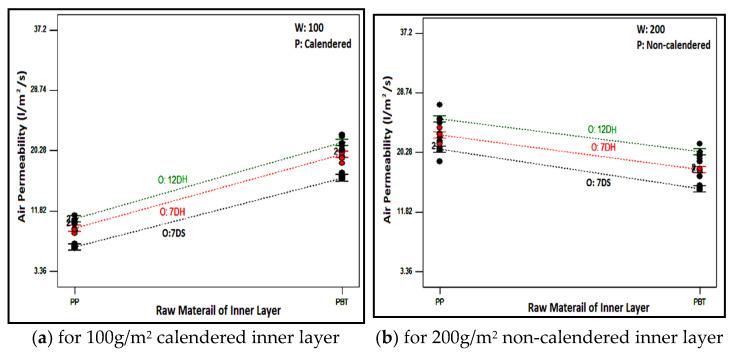
Relation between raw material of inner layer and outer layer type with composite air permeability for composites including different inner layers.

**Figure 6 polymers-16-01391-f006:**
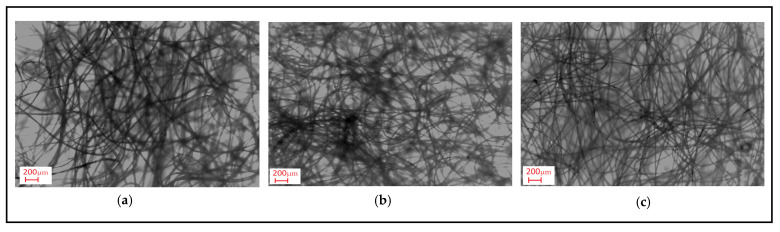
Surface image of thermo-bonded outer layer nonwovens produced from different fibers (1.5×). (**a**) 12DH fiber containing outer layer; (**b**) 7DH fiber containing outer layer; (**c**) 7DS fiber containing outer layer.

**Figure 7 polymers-16-01391-f007:**
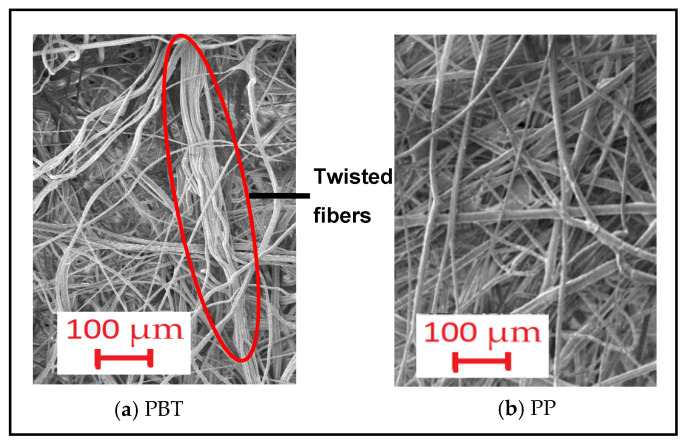
Surface image of non-calendared PBT and PP based meltblown with 200 g/m^2^ areal weight (100×).

**Figure 8 polymers-16-01391-f008:**
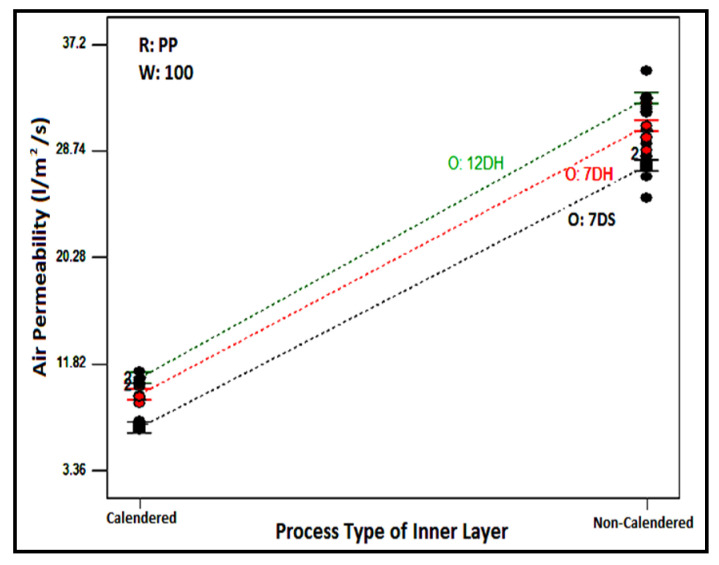
Relation between process type of inner layer and outer layer type with composite air permeability for PP based 100 g/m^2^ inner layer.

**Figure 9 polymers-16-01391-f009:**
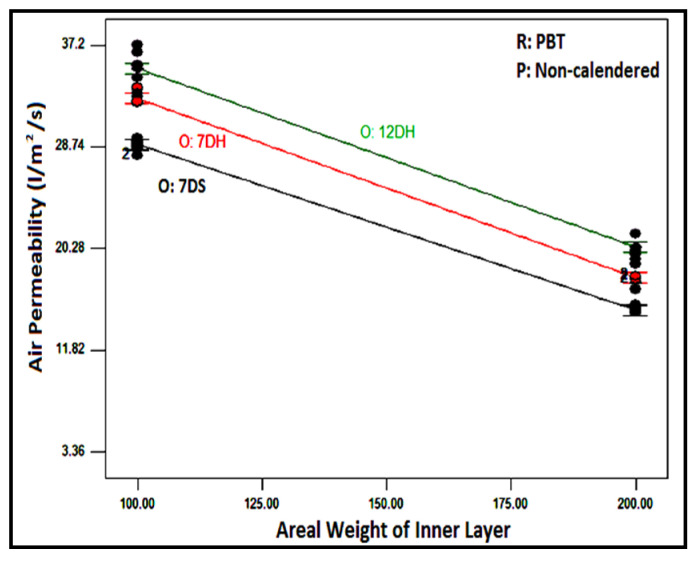
Relation between the areal weight of the inner layer and outer layer type with composite air permeability for PBT-based non-calendared inner layer.

**Figure 10 polymers-16-01391-f010:**
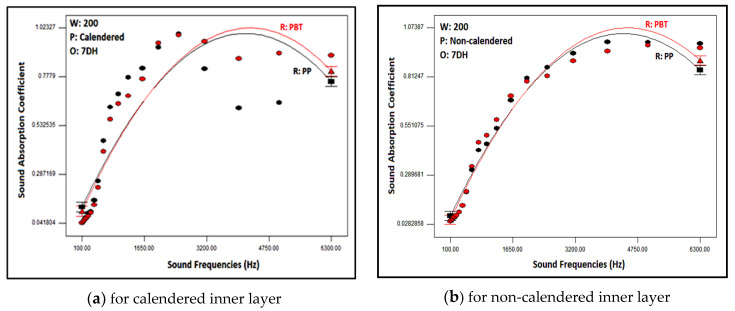
Relation between raw material of inner layer and sound frequency with composite sound absorption coefficient for composites including 200 g/m^2^ calendered and non-calendered inner layer.

**Figure 11 polymers-16-01391-f011:**
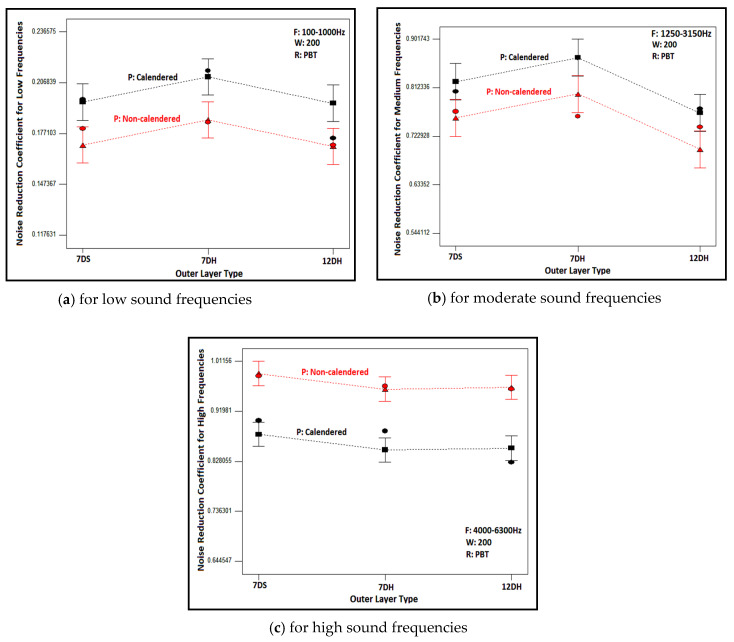
Relation between outer layer type and inner layer process type with NRC of composites containing PBT based 200 g/m^2^ areal weighted inner layer for different sound frequencies.

**Figure 12 polymers-16-01391-f012:**
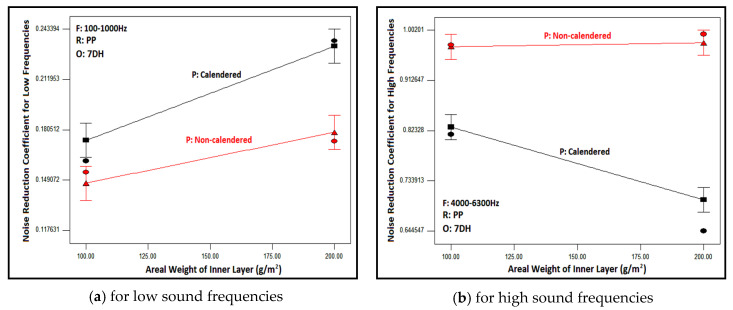
Relation between areal weight and process type of inner layer with NRC of composites containing PP-based inner layer and 7DH outer layer at different sound frequencies.

**Figure 13 polymers-16-01391-f013:**
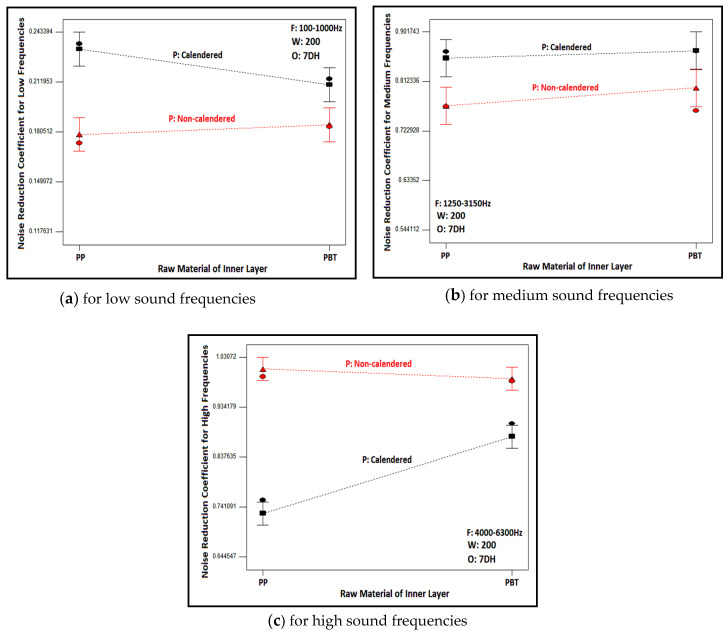
Relation between raw material and process type of inner layer with NRC of composites containing 200 g/m^2^ areal weighted inner layer and 7DH outer layer for different sound frequencies.

**Figure 14 polymers-16-01391-f014:**
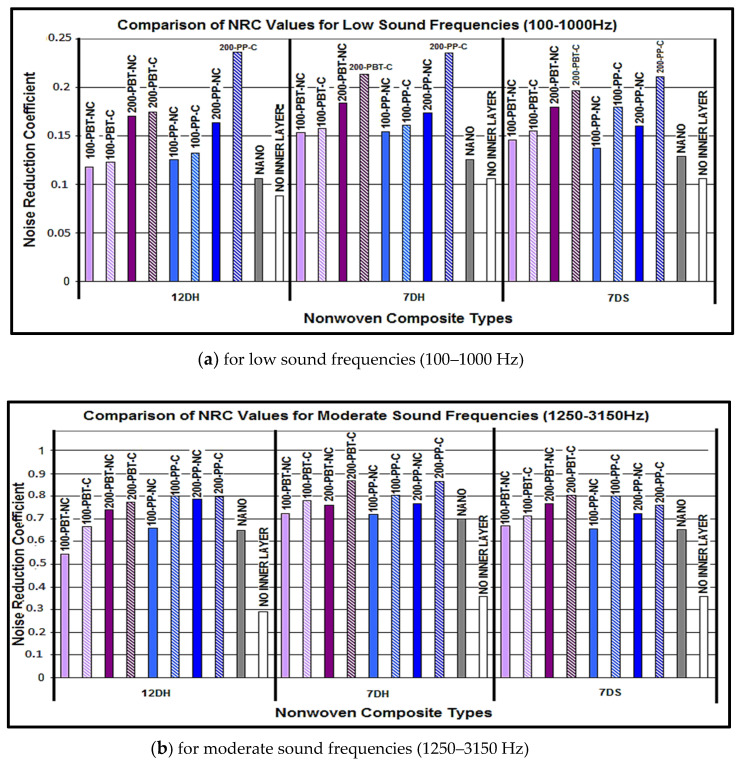
Comparison of NRC values for different nonwoven composite structures at different sound frequencies.

**Figure 15 polymers-16-01391-f015:**
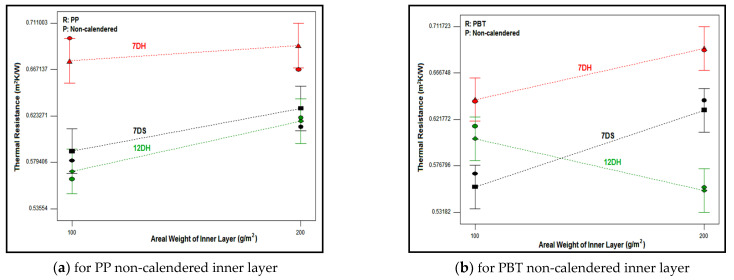
Relation between thermal resistance of composites and areal weight of inner layer for different outer layer type.

**Figure 16 polymers-16-01391-f016:**
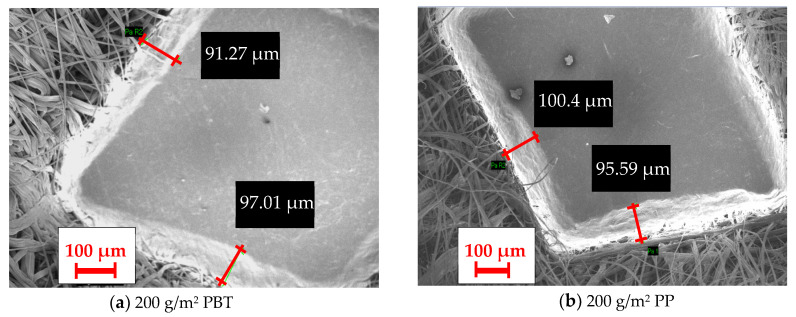
The surface image of the calendered region in different calendered inner layers (100×).

**Figure 17 polymers-16-01391-f017:**
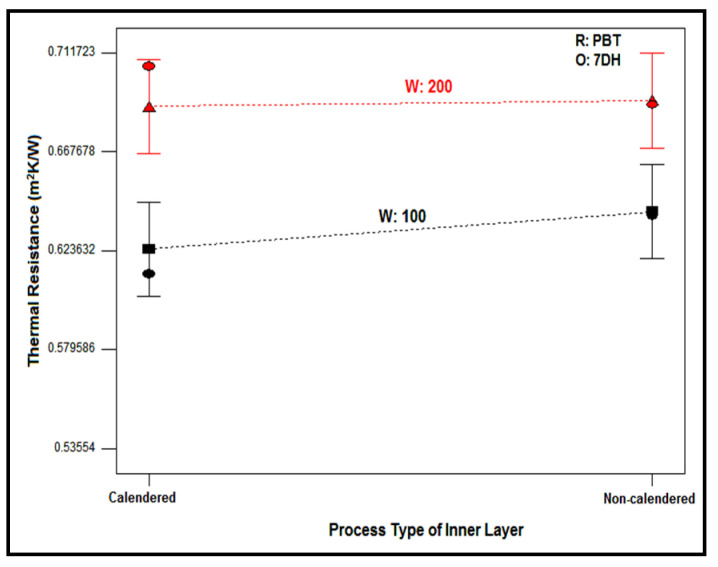
Relation between thermal resistance of composites and process types of inner layer for different areal weights and raw material type of inner layer.

**Table 1 polymers-16-01391-t001:** The properties of fibers constituting the thermo-bonded outer layers.

Fiber Type	Linear Density (denier)	Length (mm)	Strength (g/denier)	Elongation (%)	Crimp (Crimps/cm)	Cross Section (1000×)
7D Solid r-Pet (7DS)	6.8 (2.8)	64.4 (1.3)	3.6 (2.2)	63 (2.5)	3.2 (0.7)	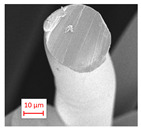
7D Hollow r-Pet (7DH)	7.3 (4.1)	64.5 (0.9)	4.01 (2.6)	42 (2.4)	-	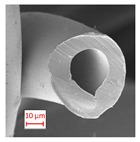
12D Hollow r-Pet (12DH)	12.2 (3.3)	64.3 (0.7)	3.55 (3.8)	60.3 (3.7)	5.1 (0.5)	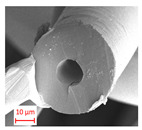
4D Bicomponent Polyester	4.2 (2.7)	51.1 (1.2)	3.68 (4.1)	54.15 (3.9)	2.75 (0.8)	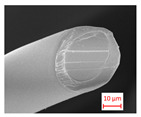

The values in parenthesis show the Coefficient of Variation (CV) % values.

**Table 2 polymers-16-01391-t002:** Thermo-bonded outer layers of composite structures.

Outer Web Layer	Content
7DS	80%–7D Solid r-PET and 20% 4D Bicomponent PET
7DH	80%–7D-Hollow r-PET and 20% 4 D Bicomponent PET
12DH	80%–12D Hollow r-PET and 20% 4 D Bicomponent PET

**Table 3 polymers-16-01391-t003:** Inner layers of composite structures.

Layer Code	Raw Material	Manufacturing Process	Target Areal Weight (g/m^2^)
PP-NC-100	Polypropylene	Produced with meltblown technology and non-calendared	100
PP-NC-200	Polypropylene	Produced with meltblown technology and non-calendared	200
PP-C-100	Polypropylene	Produced with meltblown technology and calendared	100
PP-C-200	Polypropylene	Produced with meltblown technology and calendared	200
PBT-NC-100	Polybutylene Terephthalate	Produced with meltblown technology and non-calendared	100
PBT-NC-200	Polybutylene Terephthalate	Produced with meltblown technology and non-calendared	200
PBT-C-100	Polybutylene Terephthalate	Produced with meltblown technology and calendared	100
PBT-C-200	Polybutylene Terephthalate	Produced with meltblown technology and calendared	200
N	Polyamide	Electrospun	8

**Table 4 polymers-16-01391-t004:** Properties of thermo-bonded outer layers.

Outer Web Layer	Content	Areal Weight (g/m^2^)	Thickness (mm)	Calculated Bulk Density (g/cm^3^)	Calculated Porosity (%)	Air Permeability (L/m^2^/s)
7DS	80%–7D Solid r-PET 20% 4D Bicomponent PET	200.99 (0.68)	12.74 (4.57)	0.0158	98.83	216.60 (4.5)
7DH	80%–7D-Hollow r-PET 20% 4 D Bicomponent PET	201.12 (0.55)	14.72 (4.16)	0.0137	98.99	238.00 (4.1)
12DH	80%–12D Hollow r-PET 20% 4 D Bicomponent PET	201.81 (0.50)	11.68 (3.86)	0.0173	98.72	279.40 (3.3)

The values in parenthesis show the Coefficient of Variation (CV%) values.

**Table 5 polymers-16-01391-t005:** Properties of inner layers.

Sample Code	Fiber Fineness (µm)	Areal Weight (g/m^2^)	Thickness (mm)	Strength in MD (N)	Elongation in MD (%)	Strength in CD (N)	Elongation in CD (%)	Calculated Bulk Density (g/cm^3^)	Calculated Porosity (%)	Pore Size (ìm)	Air Permeability (L/m^2^/s)
PP-C-100	3.87 (7.8)	102.1 (3.9)	0.72 (2.3)	42 (3.5)	30 (1.9)	44 (1.2)	35 (1.8)	0.159	82.344	16.5 (5.3)	4.64 (6.0)
PP-C-200	4.33 (9.3)	208.4 (3.6)	0.87 (2.5)	110 (2.8)	30 (2.3)	60 (3.1)	40 (2.2)	0.241	73.260	15.0 (4.8)	1.27 (12.6)
PP-NC-100	6.46 (8.7)	108.95 (4.3)	1.24 (4.4)	18.94 (5.1)	10.55 (3.8)	33.83 (4.2)	28.56 (4.7)	0.088	90.243	33.5 (6.2)	30.08 (5.4)
PP-NC-200	5.04 (10.3)	210.43 (5.9)	2.36 (3.8)	27.23 (4.9)	10.32 (4.2)	61.33 (3.9)	34.6 (4.1)	0.089	90.093	31.5 (5.2)	21.06 (5.9)
PBT-C-100	4.97 (8.6)	99.08 (3.9)	0.57 (1.9)	41.30 (2.4)	50.99 (3.3)	34.02 (2.9)	52.94 (2.3)	0.175	87.033	23.2 (4.8)	15.24 (6.6)
PBT-C-200	7.69 (11.2)	204.39 (2.3)	0.80 (2.4)	74.69 (2.6)	37.07 (2.8)	66.10 (2.5)	45.15 (2.4)	0.255	81.123	20.6 (4.4)	6.07 (12.3)
PBT-NC-100	4.97 (7.9)	103.97 (3.9)	0.63 (3.3)	9.20 (4.1)	43.79 (4.9)	22.59 (4.2)	38.86 (4.3)	0.164	87.853	26.2 (6.3)	30.62 (5.8)
PBT-NC-200	6.11 (8.4)	218.22 (2.9)	1.26 (2.8)	20.11 (4.7)	18.2 (3.9)	63.39 (3.8)	68.70 (3.7)	0.174	87.144	22.7 (5.9)	13.06 (9.8)
N	0.19 (10.9)	8.3 (3.3)	-	-	-	-	-	-	-	0.38 (10.2)	-

The values in parenthesis show the Coefficient of Variation (CV%) values.

**Table 6 polymers-16-01391-t006:** Summarized ANOVA for thickness and bulk density of composite structures.

ANOVA for Thickness				ANOVA for Bulk Density			
Source	F-Value	*p*-Values	Contribution (%)	Source	F-Value	*p*-Values	Contribution (%)
Model	130.079	<0.0001	R^2^ = 98.28	Model	19.01	<0.0001	R^2^ = 98.82
R	4.866	0.0424	0.53	R	1.29	0.2725	0.095
W	16.138	0.0010	1.74	W	633.85	<0.0001	46.60
O	428.750	<0.0001	92.55	O	340.14	<0.0001	50.02
P	18.992	0.0005	2.05	P	12.17	0.0030	0.89
RWO	6.529	0.0085	1.41	RP	4.90	0.0416	0.36
Residual			1.72	RWP	11.60	0.0036	0.85
Cor Total			100	Residual			1.18
				Cor Total			100

R: Raw material of inner layer, W: areal weight of inner layer, O: type of outer layer, P: process type of inner layer.

**Table 7 polymers-16-01391-t007:** ANOVA table for air permeability of composite structures.

Source	Sum of Squares	Degrees of Freedom	Contribution (%)	Mean Square	F Value	P > F	Significance
Model	10,790.99	15	R^2^ = 99.29	719.40	966.21	<0.0001	Significant
R	277.10	1	2.55	277.10	372.16	<0.0001	Significant
W	2277.93	1	20.96	2277.93	3059.44	<0.0001	Significant
O	430.25	2	3.96	215.12	288.93	<0.0001	Significant
P	6700.44	1	61.65	6700.44	8999.23	<0.0001	Significant
RW	275.76	1	2.54	275.76	370.37	<0.0001	Significant
RO	4.64	2	0.04	2.32	3.12	0.0486	Significant
RP	640.10	1	5.89	640.10	859.71	<0.0001	Significant
WO	8.01	2	0.07	4.00	5.38	0.0060	Significant
WP	161.40	1	1.49	161.40	216.78	<0.0001	Significant
OP	9.35	2	0.09	4.67	6.28	0.0027	Significant
RWP	6.03	1	0.06	6.03	8.09	0.0054	Significant
Residual	77.43	104	0.71	0.75	-	-	
Cor Total	10,868.43	119	100		-	-	

**Table 8 polymers-16-01391-t008:** ANOVA for the sound absorption coefficient of composite structures in all sound frequencies.

Source	Sum of Squares	Degrees of Freedom	Contribution (%)	Mean Square	F Value	P > F	Significance
Model	54.01	10	R^2^ = 94.79	5.40	803.6	<0.0001	Significant
R	0.0007	1	0.001	0.0007	0.1	0.7474	Not significant
W	0.16	1	0.28	0.16	23.6	<0.0001	Significant
P	0.03	1	0.05	0.03	4.3	0.0384	Significant
O	0.07	2	0.13	0.035	5.4	0.0050	Significant
F	42.45	1	74.50	42.45	6315.4	<0.0001	Significant
F^2^	11.20	1	19.66	11.20	1666.2	<0.0001	Significant
RF	0.05	1	0.08	0.05	6.8	0.0094	Significant
WF	0.09	1	0.15	0.09	12.7	0.0004	Significant
PF	0.19	1	0.34	0.19	28.6	<0.0001	Significant
Residual	2.97	442	5.21	0.007	-	-	-
Cor Total	56.98	456	100	-	-	-	-

**Table 9 polymers-16-01391-t009:** Summarized ANOVA for NRC in low, medium and high frequencies.

NRC for Low Frequencies (100–1000 Hz)				NRC for Medium Frequencies (1250–3000 Hz)				NRC for High Frequencies (4000–6300 Hz)			
Source	F Value	P > F	Contribution (%)	Source	F Value	P > F	Contribution (%)	Source	F Value	P > F	Contribution (%)
Model	21.47	<0.0001	R^2^ = 93.25	Model	16.06	<0.0001	R^2^ = 95.43	Model	47.25	<0.0001	R^2^ = 95.39
R	3.37	0.0879	1.63	R	8.33	0.0162	3.81	R	37.98	<0.0001	10.95
W	109.47	<0.0001	52.82	W	58.33	<0.0001	26.66	W	33.93	<0.0001	9.78
P	34.04	<0.0001	16.42	P	63.16	<0.0001	28.87	P	147.00	<0.0001	42.39
O	9.79	0.0022	9.44	O	16.52	0.0007	15.10	O	2.98	0.0792	1.72
RP	10.30	0.0063	4.97	RW	9.40	0.0119	4.29	RP	63.76	<0.0001	18.39
WP	9.223	0.0089	4.45	RO	7.59	0.0099	6.94	WP	42.14	<0.0001	12.15
WO	3.64	0.0533	3.51	WP	5.54	0.0404	2.53	Residual	-	-	4.61
Residual			6.75	WO	3.17	0.0858	2.89	Cor Total	-	-	100
Cor Total			100	WPO	4.74	0.0356	4.33	-	-	-	-
-	-	-	-	Residual	-	-	4.57	-	-	-	-
-	-	-	-	Cor Total	-	-	100	-	-	-	-

**Table 10 polymers-16-01391-t010:** ANOVA table of thermal resistance of nonwoven composites.

Source	Sum of Squares	Degrees of Freedom	Contribution (%)	Mean Square	F Value	P > F	Significance
Model	0.052	9	R^2^ = 88.94	0.006	12.51	<0.0001	Significant
R	0.002	1	2.87	0.002	3.64	0.0772	Not significant
W	0.004	1	6.16	0.004	7.80	0.0144	Significant
O	0.035	2	60.13	0.0175	38.06	<0.0001	Significant
P	0.0006	1	0.98	0.0006	1.23	0.2853	Not significant
RWO	0.007	2	12.65	0.0035	8.02	0.0048	Significant
WOP	0.004	2	6.15	0.002	3.89	0.0454	Significant
Residual	0.007	14	11.16	0.0005	-	-	-
Cor Total	0.059	23	100	-	-	-	-

## Data Availability

Data is provided in the article.

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
