# Peer review of "Assessing the Sound and Heat Insulation Characteristics of Layered Nonwoven Composite Structures Composed of Meltblown and Recycled Thermo-Bonded Layers"

_polymers, 2024, doi:10.3390/polym16101391_

Round 1
Reviewer 1 Report
Comments and Suggestions for Authors
(1)Try to cite reference within the last 5 years.
(2)The word of the introduction is not sufficiently concise and is too cumbersome.
(3)Confusing labelling of references, please check carefully, for example [55-81] on line 129.
(4)Punctuation error on lines 51 and 60.
(5)Is this expression correct in lines 176 %30 and %80.
(6)All of the SEM images in the paper have no scale or unclear scales.
(7)How the contribution is obtained in Table10.
Comments on the Quality of English LanguageAccording to the norms of scientific and technological English expression, the general past tense is used to describe the process of conducting experiments, and the general present tense is used to describe the conclusions (because research is a natural science, and the conclusions are objective facts), but almost all of this paper is in the past tense, which does not conform to the norms of scientific and technological English expression. The English writing skills of the author of this article need to be improved appropriately
Author Response
Dear Reviewer,
Please find our responses to your comments and criticisms in the attached document. We have revised our manuscript, taking into account your valuable feedback to the best of our ability. We appreciate the time you have dedicated to reviewing our work.
Thank you for your contributions.
With our best regards

Reviewer 2 Report
Comments and Suggestions for Authors
Overall, the author provides an in-depth experiment on composites. However, authors must carefully polish their papers.
The author needs to reduce the length, so the paper can have essential information and expressions.
2.1 Describe materials instead of process. Move processes to methods.
Sophisticate all figures to make them more understandable. This will improve the overall quality of the research.
Add scale bars and enlarge words in SEM photos.
There are many typos, such as electrospinning, nanofiber, thickness, Etc.
In tables, avoid repeated words.
Adjust the range of the vertical axis of plots. Moreover, combine plots.
What is the 'significant' in tables?
Consider combining Figure 14 into a single plot. Current expression is quite invisible to readers.
Consistency of picture and table types is necessary.
Author Response

(The authors gave the same response as above.)

Reviewer 3 Report
Comments and Suggestions for Authors
Overall, the work is well-structured, and this contribution should be considered for publication after addressing the following comments.
1. The title is too lengthy; a concise alternative might be more effective.
2. please provide the modified abstract for English correction.
3. There are too many keywords. Please specify the key keyword
4. What future research avenues do you suggest based on the findings of your study?
5. In section 2 parts 2.1 As mentioned in previous studies which studies? Need explanation and reference.
6. At the end of Table 1, "CV%" stands for "Coefficient of Variation. Write completely.
7. In section 2.2, the equation requires numbering and referencing. Please provide numbering to all equations.
8. In Table 5 how pore sizes were calculated, write in the text also.
Author Response

(The authors gave the same response as above.)

Reviewer 4 Report
Comments and Suggestions for Authors
The following points may be addressed in the revised version of the manuscript
Abstract: Use shorter sentences in Abstract. Check English. for example "As a result of this study it can be concluded that developed nonwoven composite structures that provides better sound absorption coefficients and similar thermal conductivity values compared to previous studies and materials in the market can be used as insulation material in required areas" . Such a long sentence and too many redundant words.
Introduction: is too lengthy and should be curtailed with focus on non wovens used for sound absorption. In one section, i,e page 2 of 33 References {1-84} is cited. How can we cite 84 references for one statement? The novelty part of the study should be declared in the last paragraph of the Introduction.
Methodology: In Figure 1, name the materials for the different layers. The thermobonding method and the calendaring method used may be described with schemes. In Table 3, Expansion of PBT under Raw Material is wrong in Table 3. Check the spelling and correct it .
Elucidate the ANOVA method used. What are the important parameters selected and their levels used in the experimental work.
Results and Discussion: In the discussion part the interpretation of the results appears to be speculative. The discussion is more like a report of observations. There is a disconnect between the sections. Since one cannot expect same quality of waste woven fabrics, how can we ensure consistent properties in the final product.
Conclusion: Can be presented as bullet points
Comments on the Quality of English LanguageEnglish used should be definitely improved. There are several grammatical errors, long and convoluted sentences which are to be shortened, refined and re written.
Author Response

(The authors gave the same response as above.)
